# Engineering synthetic signaling receptors to enable erythropoietin-free erythropoiesis

Aadit P. Shah [1,2,9], Kiran R. Majeti[2,9], Freja K. Ekman[1,2,3], Sridhar Selvaraj[2], Devesh Sharma[4,5], Roshani Sinha[4,5], Eric Soupene[6], Prathamesh Chati[7], Sofia E. Luna [1,2], Carsten T. Charlesworth[3], Travis McCreary[4,5], Benjamin J. Lesch[4,5], Tammy Tran[4,5], Simon N. Chu [4,5], Matthew H. Porteus [2] ✉ & M. Kyle Cromer [4,5,8] ✉

Blood transfusion plays a vital role in modern medicine, but frequent shortages occur. Ex vivo manufacturing of red blood cells (RBCs) from universal donor cells offers a potential solution, yet the high cost of recombinant cytokines remains a barrier. Erythropoietin (EPO) signaling is crucial for RBC development, and EPO is among the most expensive media components. To address this challenge, we develop highly optimized small molecule-inducible synthetic EPO receptors (synEPORs) using design-build-test cycles and genome editing. By integrating synEPOR at the endogenous *EPOR* locus in O-negative induced pluripotent stem cells, we achieve equivalent erythroid differentiation, transcriptomic changes, and hemoglobin production using small molecules compared to EPO-supplemented cultures. This approach dramatically reduces culture media costs. Our strategy not only addresses RBC production challenges but also demonstrates how protein and genome engineering can introduce precisely regulated cellular behaviors, potentially improving scalable manufacturing of a wide range of clinically relevant cell types.

Blood cell transfusion plays an essential role in modern medicine. In support of surgery, obstetrics, trauma care, and cancer chemotherapy, approximately 35,000 units of blood are drawn daily in the U.S., contributing to an annual provision of 12 million red blood cell (RBC) units[1]. However, availability is contingent on donated blood, resulting in supply constraints and safety concerns. Blood shortages pose a significant global healthcare challenge, expected to worsen with aging populations and decreasing donor numbers[2]. Moreover, patient populations with especially rare blood types constitute up to 5% of blood transfusion cases[3] and are most vulnerable to these shortages. From a financial perspective, the cost of RBC transfusion has been steadily increasing over the past two decades, accounting for nearly 10% of total inpatient hospital expenditure[4]. Collectively, these factors are expected to worsen the significant unmet medical need for transfusable blood.

To address these challenges, ex vivo manufacturing of RBCs in bioreactors from producer cell lines, such as pluripotent stem cells (PSCs), emerges as a potentially renewable and scalable solution[5]. Early clinical trials have shown that ex vivo-derived RBCs may be delivered to patients with no reported adverse events[6]. In addition, ex vivo-derived RBCs offer potential benefits compared to donor blood, including a lower risk of infectious disease transmission, streamlined production, product uniformity, and ability to source or genetically engineer antigen-negative cells[2]. However, ex vivo RBC production is

[1]School of Medicine, Stanford University, Stanford, CA, USA. [2]Department of Pediatrics, Stanford University, Stanford, CA, USA. [3]Department of Genetics, Stanford University, Stanford, CA, USA. [4]Department of Surgery, University of California, San Francisco, San Francisco, CA, USA. [5]Eli & Edythe Broad Center for Regeneration Medicine, University of California, San Francisco, San Francisco, CA, USA. [6]Benioff Children's Hospital Oakland, University of California, San Francisco, CA, USA. [7]Department of Biological & Medical Informatics, University of California, San Francisco, San Francisco, CA, USA. [8]Department of Bioengineering & Therapeutic Sciences, University of California, San Francisco, San Francisco, CA, USA. [9]These authors contributed equally: Aadit P. Shah, Kiran R. Majeti. ✉e-mail: mporteus@stanford.edu; kyle.cromer@ucsf.edu

still prohibitively expensive, owing in large part to the high cost of recombinant cytokines required to stimulate producer cells to expand and differentiate into erythroid cells[7]. Erythropoietin (EPO) signaling through the EPO receptor (EPOR) is indispensable to RBC development[8], and of all components in erythroid-promoting media, EPO is one of the most expensive[7]. Given prior success manipulating the EPOR to increase erythropoietic output[9] and the ease with which erythroid development is modeled ex vivo[10], in this work, we use synthetic biology tools and genome editing technology to de-couple EPOR signaling from the EPO cytokine.

The cellular mechanisms that regulate erythroid differentiation from hematopoietic stem and progenitor cells (HSPCs) are well understood, and efficient differentiation requires activation of the EPOR/JAK/STAT signaling cascade by EPO[11]. In its native form, two EPOR monomers dimerize in the presence of EPO to activate downstream signaling[12]. Prior work has shown that EPOR dimerization may be initiated by a range of dimer orientations and proximities using agonistic diabodies or in the context of chimeric receptors[12–14]. Because mutant FK506 binding proteins (FKBP)-based dimerization domains have been deployed to create small molecule-inducible safety switches[15], we hypothesize that FKBP domains may be repurposed to create synthetic EPOR receptors to place EPO signaling under control of a small molecule.

Here, we demonstrate that EPOR signaling can be induced by small molecule stimulation of highly optimized chimeric receptors—hereafter termed synthetic EPORs (synEPORs). We then use homology-directed repair genome editing to integrate these synEPORs under regulation of various endogenous and exogenous promoters to identify strategies that best recapitulate endogenous EPOR signaling. In this way synthetic biology is enabled by both protein engineering and precision genome engineering through the integration of full gene cassettes in a variety of genomic locations.

This work establishes synEPORs as tools that enable highly efficient ex vivo production of RBCs using a low-cost small molecule. By removing dependence on one of the most expensive elements of ex vivo erythrocyte production, these efforts address one of the major barriers to meeting the global demand for blood with ex vivo-manufactured RBCs. More broadly, this work demonstrates how synthetic biology and genome editing may be combined to introduce precisely regulated and tunable behavior into cells for a wide variety of therapeutic applications.

## Results

### FKBP-EPOR chimeras enable small molecule-dependent erythropoiesis

Perhaps the simplest genome engineering approach to replace the demand for exogenous EPO from ex vivo erythroid differentiation would be to engineer HSPCs to secrete their own EPO cytokine. To test this, we created a strategy to integrate a cassette at the *CCR5* safe harbor locus that expresses *EPO* cDNA under the strong, constitutive SFFV promoter (Supplementary Fig. 1A). We then edited HSPCs and performed ex vivo erythroid differentiation in the absence of EPO. We found that while edited cells analyzed at d14 effectively acquired erythroid markers, total cell expansion was >50-fold lower than that achieved by the addition of exogenous EPO on unedited cells (Supplementary Fig. 1B-D). In addition, we tested a commercially available EPO mimetic (EMP17) and found that it mediated little cell expansion or acquisition of erythroid markers above unedited cells cultured without EPO. This is in line with the reported low activity and specificity of EPO mimetics in comparison to recombinant EPO cytokine[16,17].

Therefore, we posited that engineering of the EPOR itself could better enable EPO-free erythropoiesis. To do so, we explored the possibility of repurposing FKBP domains to dimerize EPOR monomers and initiate downstream EPOR signaling by first designing a set of seven candidate FKBP-EPOR chimeras. We selected an FKBP-based system for dimerization due to their clinical relevance, since FKBP-Caspase9-based safety switches are currently being tested in clinical trials for their ability to eliminate engineered immune cells after transplantation (NCT01494103). Because our specific application was to enable EPO-free production of erythroid cells ex vivo, we avoid the potential for any pleiotropic effects of administering the small molecule in vivo. Therefore, we developed a variety of FKBP-EPOR chimeras that placed the FKBP domain at the N-terminus, C-terminus, at various locations within the native EPOR, and as a full replacement of the EPOR extracellular domain (Fig. 1A). DNA donor templates corresponding to each design were packaged in AAV6 vectors and integrated into the *CCR5* safe harbor site in human primary HSPCs using combined CRISPR/AAV6-mediated genome as previously described[18–20]. Expression of each FKBP-EPOR chimera was driven by the strong, constitutive SFFV promoter followed by a 2A-YFP to allow fluorescent readout of edited cells (Fig. 1A). Edited HSPCs were then subjected to an established 14-day ex vivo erythrocyte differentiation protocol[21,22] in the absence of EPO and with or without 1 nM of FKBP dimerizer AP20187 small molecule (hereafter referred to as "BB" dimerizer)[15]. Since EPO is essential for differentiation, we hypothesized that erythroid differentiation would only occur when BB stimulated a functional synEPOR to activate downstream signaling (Fig. 1B).

At the end of differentiation, we stained cells for established erythroid markers and analyzed by flow cytometry (Supplementary Fig. 2). As expected, we found that unedited "Mock" conditions yielded no erythroid cells (CD34-/CD45-/CD71+/GPA+), while HSPCs edited with synEPOR designs 1.4 and 1.5 showed BB-dependent erythroid differentiation (Fig. 1C and Supplementary Fig. 3). Although FKBP-EPOR design 1.4 appeared to be most effective, for downstream optimizations we iterated on design 1.5 due to the smaller cassette size and because removal of the entire EPOR extracellular domain is expected to eliminate potential activation by EPO cytokine. This allowed us to create a receptor that could activate the EPOR pathway only when dimerizer was present but not when endogenous cytokine was present. Further investigation of synEPOR 1.5 found a > 4x selective advantage imparted to edited cells by the end of erythroid differentiation when cells were cultured in the presence of BB without EPO as indicated by increasing edited allele frequency measured by droplet digital PCR (ddPCR) (Fig. 1D). In addition, virtually all cells that acquired erythroid markers in the synEPOR 1.5 condition were YFP+ (Fig. 1E), indicating that only edited cells were capable of differentiation.

To investigate why certain FKBP-EPOR designs were non-functional, we used AlphaFold2[23] to generate in silico structure predictions of each candidate synEPOR monomer in comparison to wild-type EPOR monomers. We observed a high-confidence structure generated across wild-type EPOR extracellular and transmembrane domains, with low-confidence scores given to signal peptide and intracellular regions (Supplementary Figs. 4, 5). For candidate synEPORs, we observed a high-confidence structure corresponding to the FKBP domain at the anticipated location among all designs. Although this analysis did not reveal any obvious protein structure disruption caused by addition of FKBP domains to the EPOR protein, our experiments demonstrated that FKBP placement within the EPOR has a great bearing on signaling potential. We found that only those constructs with FKBP placed immediately upstream of the EPOR transmembrane domain could initiate BB-dependent signaling. Therefore, it is possible that designs with FKBP within the intracellular domain may interfere with JAK/STAT signaling, while FKBPs placed further upstream of the transmembrane domain may not mediate sufficient proximity of EPOR intracellular domains to achieve sustained signaling.

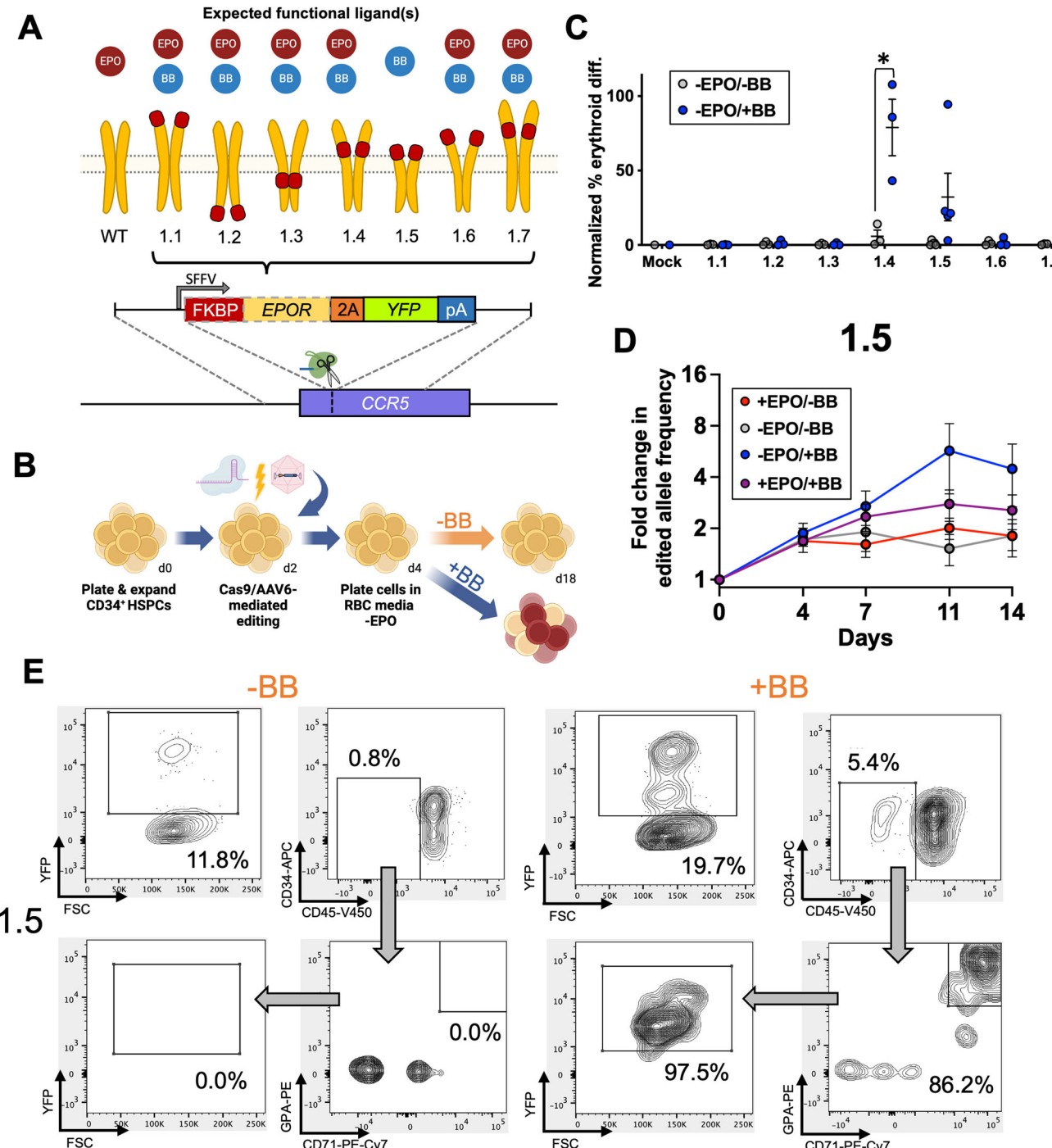

**Fig. 1 | Screening of FKBP-EPOR chimeras to facilitate EPO-free erythroid differentiation. A** Schematic of chimeric FKBP-EPOR transgenes integrated at the *CCR5* locus via CRISPR/AAV-mediated editing. Red boxes represent the location of FKBP within the EPOR. Expected functional ligands are displayed above each construct. **B** Schematic of HSPC editing and subsequent erythroid differentiation in the presence or absence of BB. Created in BioRender. Lesch, B. (2025) https://BioRender.com/k36m210. **C** Percentage of edited HSPCs that acquired erythroid markers (CD34-APC⁻/CD45-V450⁻/CD71-PE-Cy7⁺/GPA-PE⁺) +/−BB normalized to unedited cells +EPO at d14 of differentiation. Bars represent median +/−SEM; *p = 0.0196 by unpaired two-tailed *t* test across distinct samples. N = 5 biological replicates for all 1.5 conditions; N = 3 biological replicates for all 1.1, 1.2, 1.3, 1.4, 1.6, and 1.7 conditions; N = 1 biological replicate for all Mock conditions. **D** Fold change of edited allele frequencies over the course of differentiation +/−BB and +/−EPO. Bars represent median +/−SEM. N = 5 biological replicates for −BB/−EPO and +BB/−EPO conditions; N = 3 biological replicates for −BB/+EPO and +BB/+EPO conditions. Source data are provided as a Source Data file. **E** Representative flow cytometry staining and gating scheme for synEPOR 1.5-edited HSPCs at d14 of differentiation −EPO and +/−BB. Arrows indicate that only gated cells are displayed on the subsequent plot.

## Signal peptides and hypermorphic EPOR mutation increase synEPOR potency

Initial synEPOR designs 1.4 and 1.5 both mediated BB-dependent erythroid production, yet they were unable to achieve a level of differentiation equivalent to unedited cells cultured with EPO (mean of 78.9% and 32.2% of the amount of differentiation achieved with unedited cells +EPO for synEPOR 1.4 and 1.5, respectively; Fig. 1C). Therefore, we engineered second-generation synEPORs to determine if addition of a signal peptide (SP) onto synEPOR 1.5 could enhance potency, since elimination of the entire EPOR extracellular domain also

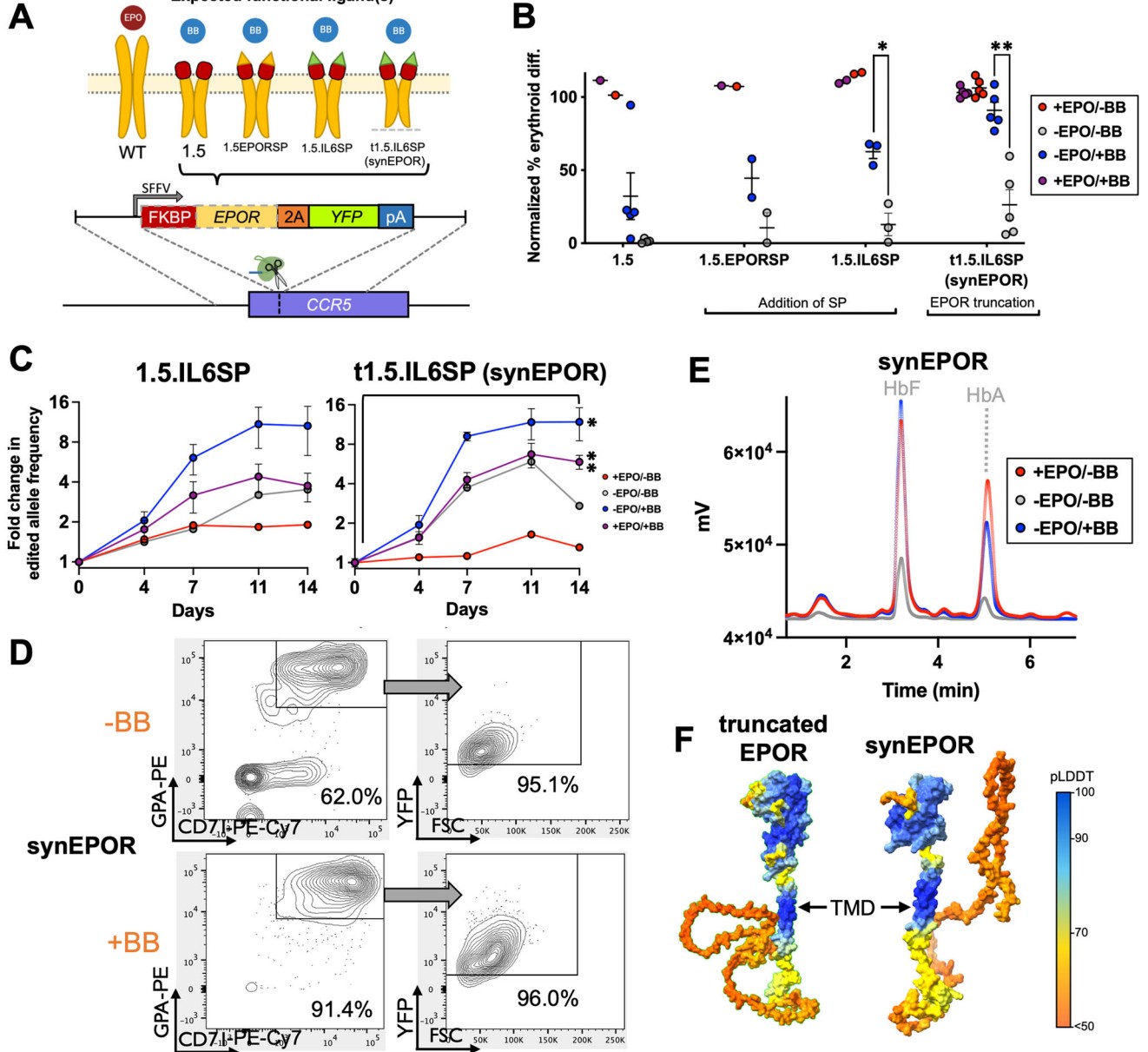

**Fig. 2 | Modulation of synEPOR effect by addition of signal peptide and EPOR truncation. A** Schematic of second-generation synEPORs integrated at *CCR5* locus. Red boxes represent the FKBP domain; yellow and green triangles indicate EPOR and IL6 SPs, respectively; dashed line represents EPOR truncation. Expected functional ligands are displayed above each construct. **B** Percentage of edited HSPCs that acquired erythroid markers (CD34-APC⁻/CD45-V450⁻/CD71-PE-Cy7⁺/ GPA-PE⁺) +/−BB normalized to unedited cells +EPO at d14 of differentiation. synE-POR 1.5 data from Fig. 1C shown for comparison. Bars represent median +/-SEM; *$p = 0.0052$ and **$p = 0.0006$ by unpaired two-tailed $t$ test across distinct samples. **C** Fold change of edited allele frequencies over the course of differentiation +/−BB and +/−EPO. Bars represent median +/−SEM; *$p = 0.0178$ and **$p = 0.000048$ comparing d0 vs. d14 within treatment by unpaired two-tailed $t$ test across distinct

samples. $N = 4$ biological replicates for all 1.5.IL6SP samples as well as −BB/−EPO and +BB/−EPO conditions edited with t1.5.IL6SP; $N = 6$ for −BB/+EPO and +BB/+EPO conditions edited with t1.5.IL6SP. **D** Representative flow cytometry staining and gating scheme for synEPOR-edited HSPCs at d14 of differentiation −EPO and +/−BB. Arrows indicate that only gated cells are displayed on the subsequent plot. **E** Representative hemoglobin tetramer HPLC plots at d14 of erythroid differentiation. +BB and −BB/−EPO conditions were from cells edited with synEPOR; +EPO condition was from unedited cells. All plots normalized to 1e6 cells. Source data are provided as a Source Data file. **F** AlphaFold2-based structure prediction of truncated EPOR and synEPOR. SP was removed since this sequence will be cleaved following translocation to the membrane. TMD labeled with an arrow as a reference point.

removes the native SP at the N-terminus. To test the effect of these modifications, we designed and built constructs that added the native EPOR SP or the IL6 SP[24] onto the N-terminus of synEPOR 1.5. This comparison was performed because SPs for cytokines are known to be particularly strong[25,26]. These DNA donor templates were packaged into AAV6 and integrated into the *CCR5* locus as before (Fig. 2A). We then performed ex vivo erythroid differentiation in the presence or absence of EPO and BB. We found that addition of EPOR SP and IL6 SP

both improved mean erythroid differentiation in the presence of BB alone (44.5% and 62.6%, respectively) compared to the original synE-POR 1.5 design (32.2%) (Fig. 2B and Supplementary Fig. 6). These vectors also yielded a further selective advantage in the presence of BB (both with and without EPO), achieving a mean 10.0- and 10.6-fold increase in edited allele frequency by the end of erythroid differentiation with addition of EPOR and IL6 SPs, respectively (Fig. 2C and Supplementary Fig. 7A).

Given the greater efficacy of synEPOR 1.5 with IL6 SP, we investigated whether incorporation of a naturally occurring nonsense mutation (*EPOR*[W439X]) that truncates the 70 C-terminal amino acids of EPOR and eliminates a negative inhibitory domain may additionally increase receptor potency[9]. Therefore, we designed a vector with this truncated EPOR intracellular domain as well as IL6 SP and observed a further enhancement, achieving a mean of 90.9% erythroid differentiation compared to EPO-cultured HSPCs (Fig. 2B). This significantly increased the selective advantage of edited cells cultured in the presence of BB, achieving a mean 11.9-fold increase in edited alleles by the end of erythroid differentiation (Fig. 2C). As before, virtually all cells that acquired erythroid markers were YFP[+], indicating that only edited cells stimulated with BB were able to initiate EPOR signaling (Fig. 2D). Notably, a substantial portion of cells also differentiated in the absence of BB, which we addressed in downstream experiments. We will hereafter refer to our optimized FKBP-EPOR design 1.5 with IL6 SP and naturally occurring EPOR truncation as "synEPOR".

To ensure that synEPOR-stimulated erythroid cells produce functional hemoglobin, we performed hemoglobin tetramer high-performance liquid chromatography (HPLC) at the end of erythroid differentiation. We found that cells edited with the optimized synEPOR and cultured with BB yielded a hemoglobin production profile consisting primarily of adult and fetal hemoglobin (HbA and HbF, respectively). This hemoglobin production profile was indistinguishable from that produced by unedited cells culture with EPO (Fig. 2E).

Finally, we used AlphaFold2 to predict the structure of the optimized synEPOR and find remarkable similarity to the predicted structure of the naturally occurring truncated EPOR (Fig. 2F). As expected, we observe a shortening of the low-confidence intracellular domain for the truncated EPOR compared to wild-type EPOR (Supplementary Fig. 8) as well as a high-confidence structure corresponding to the FKBP domain in the expected location for the optimized synEPOR. As with native EPOR SP, we also observe a low-confidence region corresponding to the IL6 SP.

## Genome engineering modulates synEPOR function

While our optimized synEPOR was effective at mediating small molecule-dependent erythroid differentiation and hemoglobin production, we observed some erythroid differentiation and hemoglobin production in the absence of BB as well (Fig. 2B, E). This could be due to the strong, constitutive viral SFFV promoter driving supraphysiologic levels of receptor expression that induced ligand-independent dimerization. In contrast to the potent SFFV promoter, prior work has shown that CD34[+] HSPCs express low levels of the endogenous *EPOR*, and expression increases modestly over the course of ex vivo erythroid differentiation[27]. Therefore, in the next round of optimizations, we explored the impact of various expression profiles on synEPOR activity. To do so, we developed targeted integration strategies that placed an identical optimized synEPOR under expression of: 1) an exogenous yet weaker, constitutive human *PGK1* promoter following integration at the *CCR5* locus (hereafter referred to as "*PGK*(synEPOR)"); 2) the strong erythroid-specific *HBA1* promoter following integration into the start codon of the *HBA1* locus[10] (hereafter referred to as "*HBA1*(synEPOR)"); and 3) the endogenous *EPOR* locus following integration into the 3′ end of the gene and linked by a 2A cleavage peptide (hereafter referred to as "*EPOR*(synEPOR)") (Fig. 3A). We chose these additional integration strategies to investigate whether extremely high *synEPOR* expression or simply constitutive expression throughout differentiation was most responsible for the dimerizer-independent activity of SFFV(synEPOR). These experiments also investigated whether erythroid-specific expression of *synEPOR* from the highly expressed *HBA1* locus may elicit the most dramatic pro-erythroid effect or if, alternatively, integration of synEPOR at the endogenous *EPOR* locus may best recapitulate endogenous EPOR signaling—

analogous to the effective regulation of synthetic T cell receptors when knocked into the endogenous TRAC locus[28].

Following the integration of each vector into the intended site in primary HSPCs, we performed ex vivo erythroid differentiation in the presence or absence of EPO and BB. We observed that all three integration strategies yielded effective erythroid differentiation in the presence of BB compared to unedited cells cultured with EPO (Fig. 3B, C). However, the greatest differences were found in edited conditions cultured without BB or EPO. Compared to the mean 26.3% erythroid differentiation we observed previously in the SFFV(synEPOR)-edited condition without EPO or BB, expression of *synEPOR* from the *PGK* and *EPOR* promoters both reduced BB-independent activity (mean of 2.2% and 20.0% in *PGK*(synEPOR) and *EPOR*(synEPOR) conditions, respectively) (Fig. 3B, C and Supplementary Fig. 7B). In contrast, we found that expression of *synEPOR* from the *HBA1* promoter drove high frequencies of erythroid differentiation in the presence and absence of BB, indicating constitutive activity (Fig. 3B, C). Because *HBA1* is expressed much more highly than *EPOR* by the end of ex vivo differentiation[29], it is possible that this BB-independent activity is a result of supraphysiologic levels of *synEPOR* expression from the *HBA1* promoter that leads to spontaneous signaling even in the absence of dimerizing ligand. However, further experiments would be required to determine whether this is the case. Interestingly, deeper characterization of synEPOR-edited cells by d14 of differentiation revealed that *HBA1*(synEPOR) conditions yielded a less mature phenotype, with a higher percentage of CD36[+] cells (a marker of immature erythroblasts)[30] (Supplementary Fig. 9) and significantly fewer enucleated erythrocytes (Supplementary Fig. 10). On the other hand, we found that *PGK*(synEPOR) and *EPOR*(synEPOR) conditions yielded the opposite—fewer CD36[+] cells and a higher percentage of enucleated erythroblasts by d14 of differentiation. Cresyl blue staining confirmed these findings, while also indicating that the vast majority of cells in all conditions had reached terminal differentiation, becoming either normoblasts, reticulocytes, or enucleated erythrocytes (Supplementary Fig. 11).

In spite of these subtle differences, we found that each synEPOR integration strategy yielded normal production of adult and fetal hemoglobin when edited cells were cultured in the presence of BB without EPO (Fig. 3D). Due to the high level of erythroid differentiation observed in the *HBA1*(synEPOR) condition, it was unsurprising that edited cells cultured with neither EPO nor BB also produced a substantial amount of adult and fetal hemoglobin. Finally, we found that total erythroid cell production from all synEPOR-edited HSPCs cultured with BB was comparable to HSPCs cultured with exogenous EPO (Fig. 3E).

Next, we determined whether expression of *synEPOR* from these different promoters has a bearing on the dose response to BB. While prior work using BB found 1 nM to be most effective at activating small molecule-inducible safety switches[15], we observed substantial erythroid differentiation at levels well below 1 nM of BB. In fact, we found that 1pM and 10pM of BB yielded erythroid differentiation that was comparable to EPO in cells edited with *EPOR*(synEPOR) and *PGK*(synEPOR) strategies, respectively (Fig. 3F). However, to achieve mean differentiation that was identical to or greater than EPO-stimulated cells required a dose of 0.1 nM for *PGK*(synEPOR)- and *EPOR*(synEPOR)-edited populations. In contrast, we found that cells edited with *HBA1*(synEPOR) yielded efficient erythroid differentiation across the entire dose range, including in the absence of BB (Fig. 3F), consistent with constitutive activity of this integration strategy.

## synEPOR closely replicates endogenous EPOR signaling

In its native form, EPO cytokine dimerizes two EPOR monomers, leading to a JAK/STAT signaling cascade culminating in translocation of phosphorylated STAT5 to the nucleus, which initiates a pro-erythroid transcriptional program[11]. While we have shown that synEPOR-edited cells stimulated with BB acquire classic erythroid

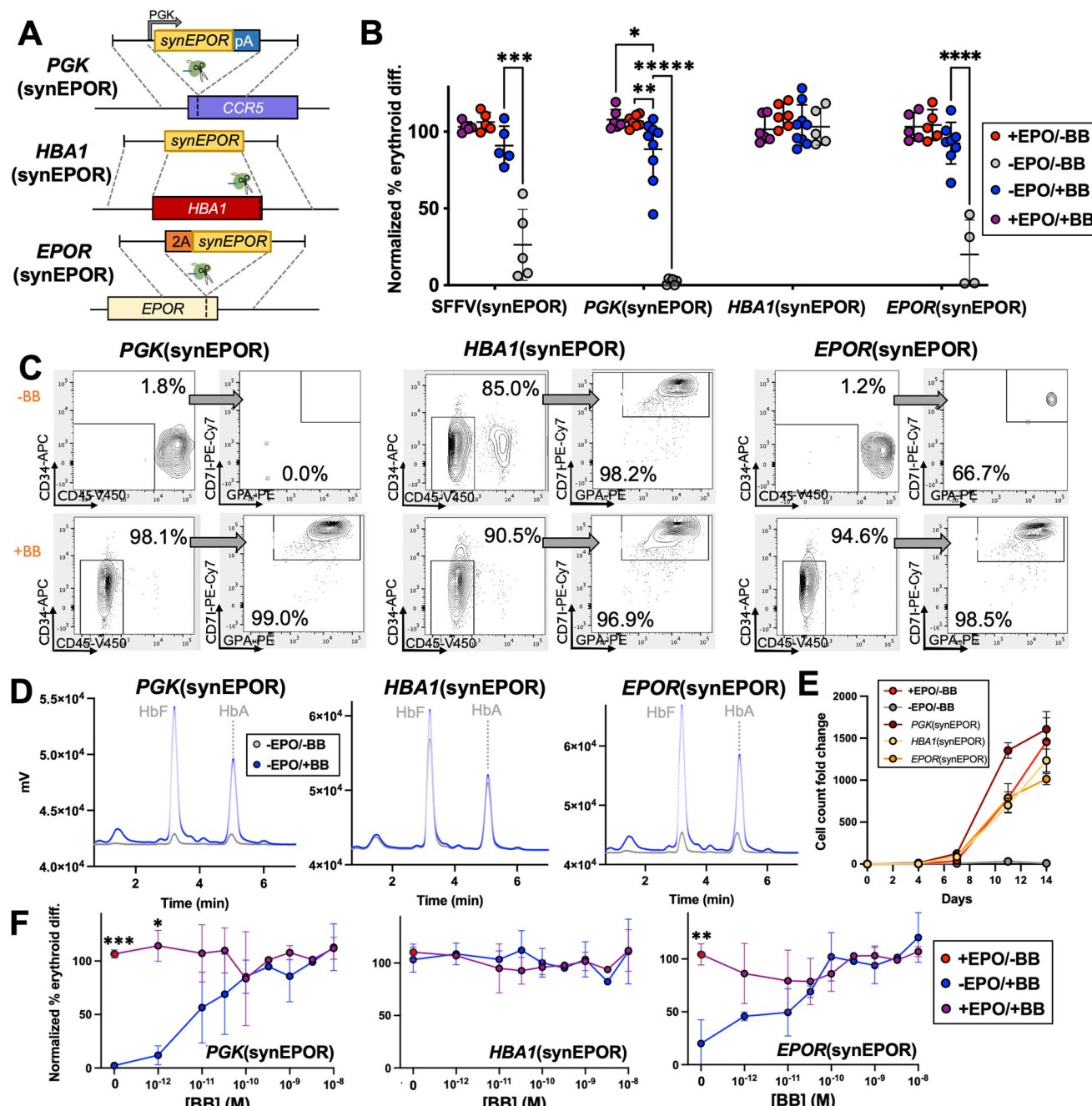

**Fig. 3 | Modulation of synEPOR effect by genome engineering. A** Schematic of third-generation synEPORs that drive expression from: (1) *PGK* promoter from *CCR5* safe harbor site; (2) erythroid-specific *HBA1* locus; and (3) endogenous *EPOR* locus. **B** Percentage of edited HSPCs that acquired erythroid markers (CD34-APC/CD45-V450⁻/CD71-PE-Cy7⁺/GPA-PE⁺) +/-BB normalized to unedited cells +EPO at d14 of differentiation. SFFV(synEPOR) data from Fig. 2B shown here for comparison. Bars represent median +/-SEM; *p = 0.0434, **p = 0.0354, ***p = 0.0006, ****p = 0.000034, and *****p < 0.00001 by unpaired two-tailed t test across distinct samples. **C** Representative flow cytometry staining and gating scheme for edited HSPCs at d14 of differentiation −EPO and +/−BB. Arrows indicate that only gated cells are displayed on the subsequent plot. **D** Representative hemoglobin tetramer HPLC plot of edited HSPCs at d14 of differentiation -EPO and +/−BB. **E** Cumulative cell count fold change of edited HSPCs over the course of differentiation. Bars represent median +/−SEM. N = 6 biological replicates for all Mock conditions; and N = 3 biological replicates for all synEPOR-edited conditions. **F** Dose response of edited HSPCs cultured over a range of [BB] at d14 of differentiation normalized to unedited cells +EPO. Bars represent median +/−SEM; *p = 0.000481, **p = 0.000126, and ***p < 0.00001 comparing +BB/+EPO to +BB/-EPO conditions by unpaired two-tailed t test across distinct samples. N = 7 biological replicates for *PGK*(synEPOR) condition +EPO at 0 M [BB];

N = 6 biological replicates for *PGK*(synEPOR) conditions +EPO at 10⁻⁹M [BB], −EPO at 0 M [BB], and −EPO at 10⁻⁹M [BB], *HBA1*(synEPOR) conditions +EPO at 0 M [BB], +EPO at 10⁻⁹M [BB], and -EPO at 10⁻⁹M [BB]; N = 5 biological replicates for *HBA1*(synEPOR) condition -EPO at 0 M [BB], *EPOR*(synEPOR) conditions +EPO at 0 M [BB], +EPO at 10⁻⁹M [BB], and −EPO at 10⁻⁹M [BB]; N = 4 biological replicates for *PGK*(synEPOR) condition +EPO at 10⁻¹⁰M [BB], *HBA1*(synEPOR) condition +EPO at 10⁻¹⁰M [BB], -EPO at 10⁻¹⁰M [BB], *EPOR*(synEPOR) condition −EPO at 0 M [BB]; N = 3 biological replicates for *PGK*(synEPOR) conditions +EPO at 10⁻¹²M [BB], +EPO at 10⁻¹¹M [BB], +EPO at 30⁻¹¹M [BB], −EPO at 10⁻¹²M [BB], -EPO at 10⁻¹¹M [BB], and −EPO at 10⁻¹⁰M [BB], *HBA1*(synEPOR) conditions +EPO at 10⁻¹²M [BB], +EPO at 10⁻¹¹M [BB], +EPO at 30⁻¹¹M [BB], -EPO at 10⁻¹²M [BB], −EPO at 10⁻¹¹M [BB], and −EPO at 30⁻¹¹M [BB], *EPOR*(synEPOR) conditions +EPO at 10⁻¹⁰M [BB], and −EPO at 10⁻¹⁰M [BB]; N = 2 biological replicates for *PGK*(synEPOR) condition +EPO at 10⁻⁸M [BB], −EPO at 30⁻¹¹M [BB], and -EPO at 10⁻⁸M [BB], *HBA1*(synEPOR) conditions +EPO at 10⁻⁸M [BB], and −EPO at 10⁻⁸M [BB], *EPOR*(synEPOR) conditions +EPO at 10⁻¹²M [BB], +EPO at 10⁻¹¹M [BB], +EPO at 30⁻¹¹M [BB], +EPO at 10⁻⁸M [BB], −EPO at 10⁻¹²M [BB], -EPO at 10⁻¹¹M [BB], −EPO at 30⁻¹¹M [BB], −EPO at 10⁻⁸M [BB]; and N = 1 biological replicate for all conditions at 30⁻¹⁰M [BB], 30⁻⁹M [BB], and 30⁻¹⁰M [BB]. Source data are provided as a Source Data file.

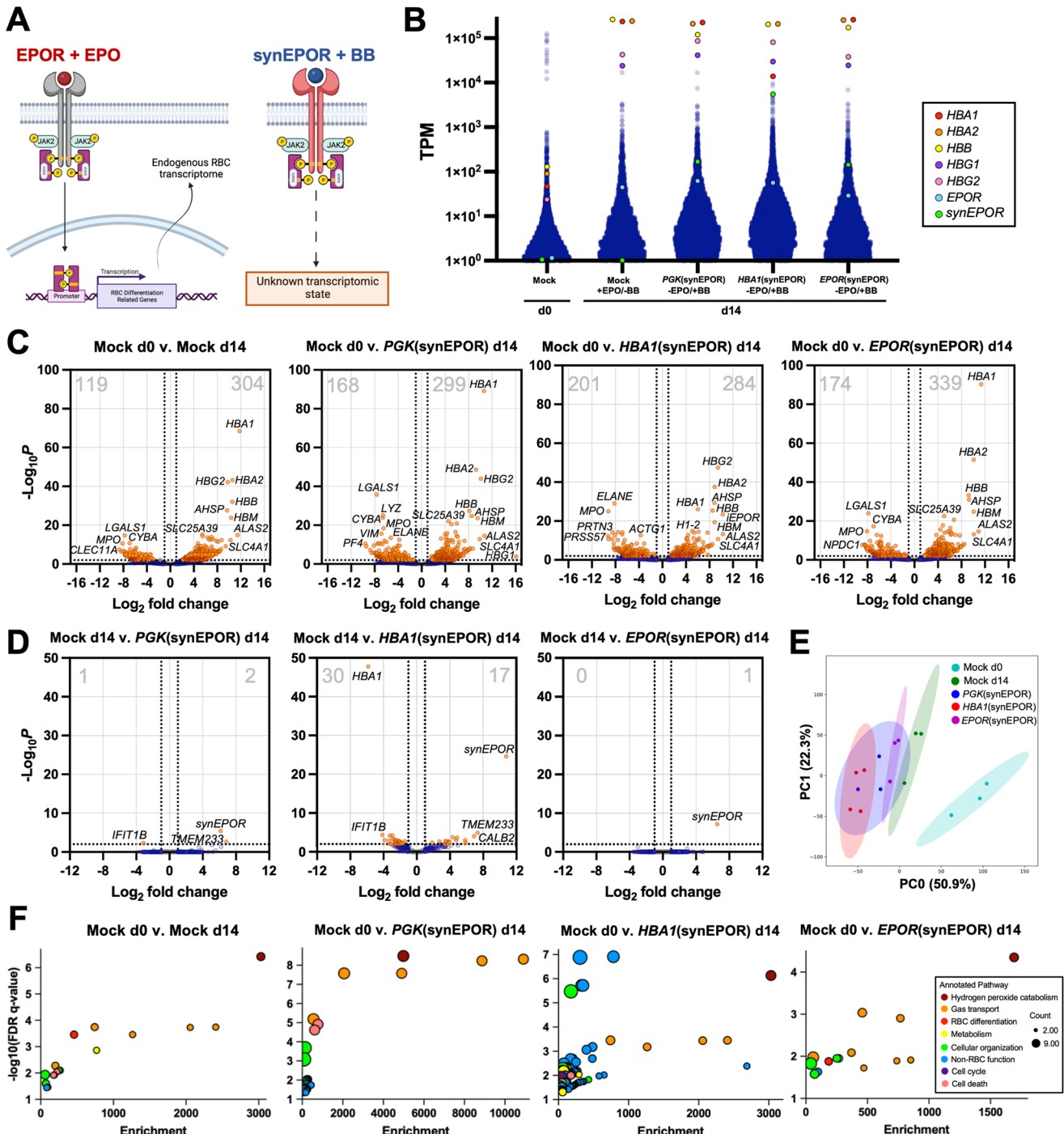

**Fig. 4 | Transcriptome-wide analysis of synEPOR-edited cells. A** Schematic of well-characterized endogenous EPO + EPOR signaling effects vs. undefined BB +synEPOR signaling effects. Created in BioRender. Lesch, B. (2025) https://BioRender.com/z58z349. **B** Transcripts per million (TPM) from RNA-Seq with annotations for globin, *EPOR*, and *synEPOR* genes. **C** Volcano plot comparing unedited and edited HSPCs at d14 of differentiation v. unedited HSPCs at d0. Dashed lines are drawn at +/−1 log$_2$ fold change and adjusted *p*-value = 0.01 by Wald test. Total number of significantly down- and upregulated genes is shown in top left and top right of each plot, respectively. **D** Volcano plot comparing edited HSPCs at d14 +BB v. unedited HSPCs at d14 +EPO. Dashed lines are drawn at +/−1 log$_2$ fold

change and adjusted *p*-value = 0.01 by Wald test. Total number of significantly down- and upregulated genes is shown in top left and top right of each plot, respectively. **E** Principal component analysis of all conditions with covariance ellipses. **F** Summary of gene ontology (GO) enrichment analysis comparing all d14 conditions v. d0 control. Top 50 differentially expressed genes were used as input. Plotted are significantly enriched GO pathways (Benjamini-Hochberg False Discovery Rate adjusted *p*-value ≤ 0.05) that were binned into broader categories with enrichment score derived by Enrichr software. Count refers to the number of genes within each GO pathway that contributed to enrichment. Source data are provided as a Source Data file.

markers and yield normal hemoglobin profiles, an open question is whether this synthetic stimulus recapitulates the complex transcriptional response of endogenous EPOR signaling (Fig. 4A). To investigate this, we edited HSPCs with our various synEPOR integration strategies

(Fig. 3A) and performed bulk RNA-sequencing (RNA-Seq) at d14 of erythroid differentiation in absence of EPO and presence of BB. For comparison, we also performed RNA-Seq on unedited cells at the beginning (d0) and end (d14) of erythroid differentiation in the

presence of EPO. These efforts yielded an average of 55.1 M reads per sample with 98.5% of reads aligned to the genome and 97.2% with Quality Score ≥20 (Supplementary Fig. 12).

In analyzing these data, we found that alpha-, gamma-, and beta-globin are among the most significantly upregulated genes in unedited cells at d14 vs. d0 (Fig. 4B, C). Similarly, for all cells edited with synEPOR, these globins are also among the most significantly upregulated genes (Fig. 4B, C). In fact, by the end of differentiation these globins comprise a mean of 86.5%, 76.9%, 63.3%, and 83.7% of all reads for unedited cells cultured with EPO as well as *PGK*(synEPOR)-, *HBA1*(synEPOR), and *EPOR*(synEPOR)-edited cells cultured with BB, respectively (Supplementary Fig. 13A and Supplementary Data 1). As previously observed, we found that *EPOR* expression increases over the course of erythroid differentiation[27] (38.8-fold from d0 to d14 in unedited cells; Fig. 4B and Supplementary Fig. 13B). In all synEPOR-edited cells we observe similar levels of *EPOR* compared to unedited cells, which is expected since each integration strategy preserves endogenous *EPOR* expression. As for *synEPOR* expression, we find that both *PGK* and *EPOR* promoters drive expression comparable to that of endogenous *EPOR* at d14 in unedited cells (Fig. 4B and Supplementary Fig. 13C). However, the *HBA1* promoter drives supraphysiologic levels of *synEPOR*, with expression approaching that of the globins. Since the *HBA1*(synEPOR) integration strategy replaces a full copy of the *HBA1* gene with *synEPOR* transgene, it is not surprising to find a significant decrease in *HBA1* expression in this condition as well (Fig. 4B, Supplementary Fig. 13A, and S13D). Consistent across donors, genes most highly expressed in HSPCs are uniformly downregulated in all d14 samples while erythroid-specific genes are uniformly upregulated (Fig. 4B, C, Supplementary Fig. 13D–H, and Supplementary Data 2). Because of this, we find that d0 HSPCs and all d14 samples segregate into two distinct hierarchies (Supplementary Fig. 14A), indicating a high degree of similarity across all d14 samples regardless of whether these were unedited cells cultured with EPO or synEPOR-edited cells cultured with BB.

Although consistent differences were observed comparing all conditions to d0 HSPCs, we next determined whether significant differences existed at the end of differentiation between unedited cells cultured with EPO and synEPOR-edited conditions cultured with BB. This comparison revealed an extremely high degree of similarity between unedited cells cultured with EPO and *PGK*(synEPOR)-edited cells cultured with BB; only three genes were differentially expressed, including upregulation of the *synEPOR* transgene (Fig. 4D). In contrast, the transcriptome of *HBA1*(synEPOR)-edited cells departed more substantially from unedited cells, with a total of 47 differentially expressed genes. As expected, in this condition we observed significant upregulation of *synEPOR* as well as downregulation of *HBA1*. Remarkably, we find that the only differentially expressed gene in *EPOR*(synEPOR)-edited conditions is the *synEPOR* transgene, indicating that this condition best recapitulated endogenous EPOR signaling. These conclusions were further supported by principal component analysis, which found that all d14 samples clustered separately from d0 samples and that *EPOR*(synEPOR)-edited cells stimulated with BB most closely resemble unedited cells cultured with EPO (Fig. 4E). Gene co-expression network analysis additionally revealed a high degree of similarity between synEPOR-edited conditions and unedited cells cultured with EPO (Supplementary Fig. 14B).

To determine which cellular processes were activated by EPO compared to edited cells cultured with BB, we performed gene ontology enrichment analysis of differentially expressed genes in each condition compared to unedited cells at d0 (Fig. 4F and Supplementary Data 3). At d14, the most highly enriched pathways were hydrogen peroxide ($H_2O_2$) catabolism—a critical function of erythrocytes to process the significant amounts of superoxide and $H_2O_2$ that occur during oxygen transport[31]. We also find gas transport and erythroid differentiation processes to be highly enriched across all d14 samples.

From this analysis, the *HBA1*(synEPOR) condition shows the most substantial departure from endogenous EPOR signaling, with a number of significantly enriched pathways unrelated to erythrocyte function. On the other hand, we find that *EPOR*(synEPOR) most closely resembles endogenous EPOR signaling, leading us to conclude that expression of synthetic receptors from the endogenous promoter is likely to best recapitulate the transcriptomic changes initiated by native cytokine signaling.

## synEPOR enables EPO-free erythropoiesis from induced pluripotent stem cells

All prior work was done in primary HSPCs to determine whether we could successfully engineer small molecule-inducible EPORs that recapitulate native erythroid development and function. However, while primary hematopoietic HSPCs may be sourced from umbilical cord blood and mobilized peripheral blood to produce RBCs ex vivo, their expansion capacity is limited[2]. As a solution, induced pluripotent stem cell (iPSC) producer lines provide a potentially unlimited source of patient-derived RBCs[6]. Therefore, in downstream experiments we used an iPSC line called PB005 derived from a healthy donor with O-negative blood type[32] to determine if synEPORs could effectively produce erythroid cells from a universal blood donor.

To test this, we integrated our most effective synEPOR expression strategies—*PGK*(synEPOR) and *EPOR*(synEPOR)—into the PB005 iPSC line and isolated homozygous knock-in clones (Fig. 5A and Supplementary Fig. 15A). These clones were then subjected to an established 12-day differentiation into hematopoietic progenitor cells (HPCs). Surprisingly, we found that *EPOR*(synEPOR)-edited clones yielded a substantially greater number of CD34[+] cells compared to both unedited and *PGK*(synEPOR)-edited clones (Supplementary Fig. 15B–D), although this condition had a higher proportion of cells staining for erythroid markers (Supplementary Fig. 15E, F). Following iPSC-to-HPC differentiation, we performed a 14-day RBC differentiation without EPO and +/−BB (Fig. 5A). We found that all cells (unedited and edited) effectively differentiated in the presence of EPO, whereas virtually no erythroid differentiation was observed in unedited cells in the absence of EPO (Fig. 5B and Supplementary Fig. 16A). *PGK*(synEPOR)-edited cells stimulated with BB yielded a high percentage of erythroid cells, but differentiation efficiency was significantly less than that achieved by EPO in these clones at every timepoint (Fig. 5B). In addition, overall cell proliferation was substantially lower than that achieved with EPO (Fig. 5C). In contrast, *EPOR*(synEPOR) clones achieved a differentiation efficiency that was indistinguishable from clones cultured with EPO; cell proliferation over the course of differentiation was also nearly equivalent to that achieved with EPO (Fig. 5B, C). Given frequent clonal differences observed in proliferation capacity, we also examined cell proliferation from the best *PGK*(synEPOR)- and *EPOR*(synEPOR)-edited clones. By day 14, the most highly proliferative *PGK*(synEPOR)-edited clone only achieved 37.3% of the proliferation of that same clone when cultured with EPO (Supplementary Fig. 16B). However, the most effective *EPOR*(synEPOR)-edited clone achieved even greater proliferation (107.8%) compared to the same clone cultured with EPO. We note that this proliferation rate was normalized to the number of HPCs at the beginning of erythroid differentiation and therefore does not take into account the increase in CD34[+] HPCs achieved within the *EPOR*(synEPOR)-edited condition over the course of iPSC-to-HPC differentiation (Supplementary Fig. 15B–D).

Next, we measured the hemoglobin profiles of these iPSC-derived erythroid cells using HPLC and observed fetal hemoglobin to be the most prevalent tetramer in the presence of EPO, which is consistent with prior studies[33]. We found this to be the case as well for clones edited with both *PGK*(synEPOR) and *EPOR*(synEPOR) conditions cultured with BB (Fig. 5D). While *PGK*(synEPOR) conditions cultured with BB almost uniformly expressed lower fetal hemoglobin than their EPO-cultured counterparts, we observed the opposite for *EPOR*(synEPOR)-

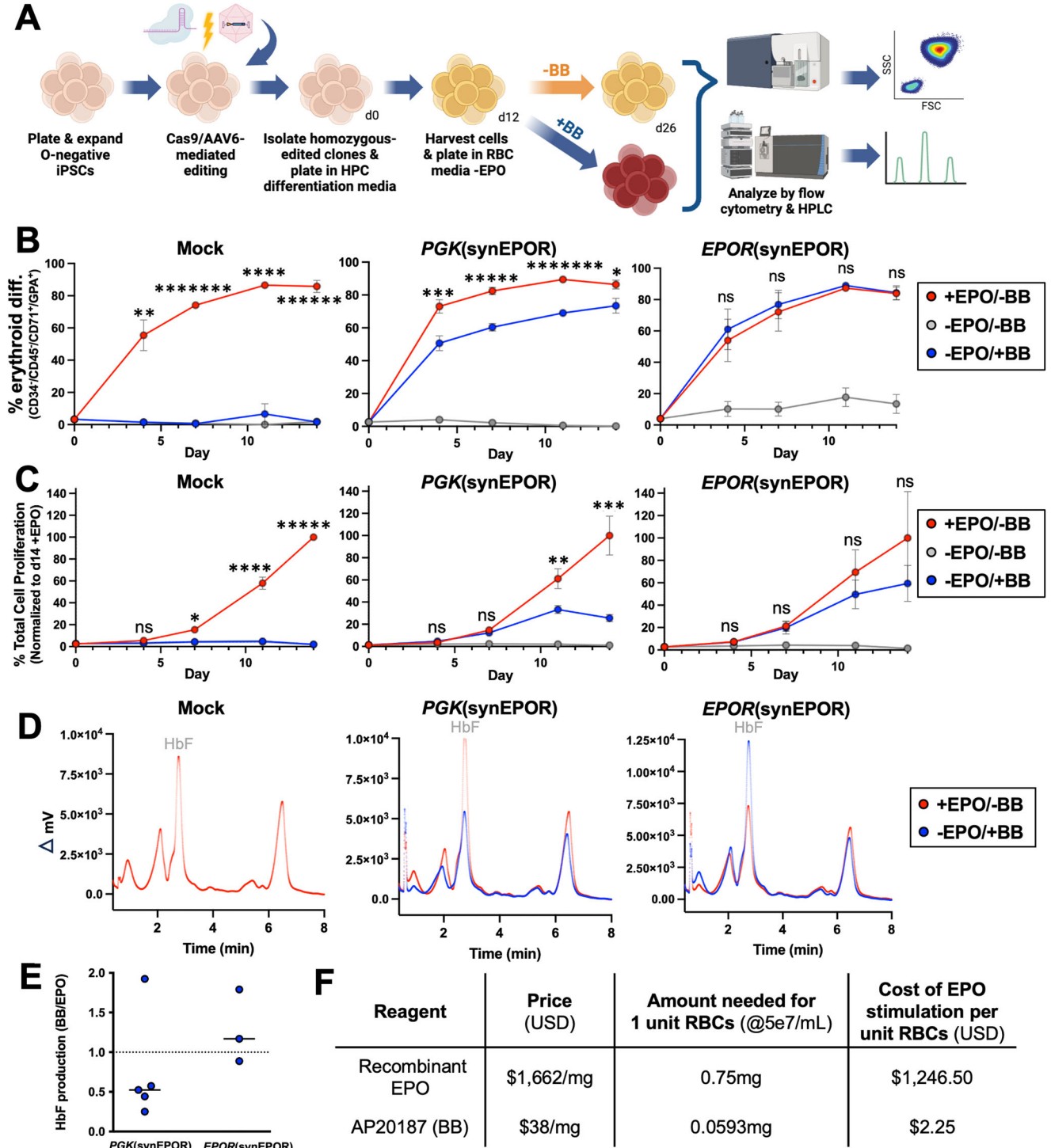

**Fig. 5 | Differentiation of iPSCs into erythroid cells using synEPOR + BB.**
**A** Schematic of iPSC-to-erythroid cell differentiation strategy and subsequent analysis. Created in BioRender. Lesch, B. (2025) https://BioRender.com/e82e687.
**B** Percentage of cells that acquired erythroid markers (CD34-APC⁻/CD45-V450⁻/CD71-PE-Cy7⁺/GPA-PE⁺) over the course of differentiation. Bars represent mean +/−SEM; ns = not statistically significant, *$p$ = 0.0353, **$p$ = 0.00469, ***$p$ = 0.00444, ****p = 0.000236, *****$p$ = 0.00004, ******: $p$ = 0.000034, and *******$p$ < 0.00001 comparing −BB/+ EPO to +BB/-EPO conditions by unpaired two-tailed $t$ test across distinct samples. $N$ = 6 biological replicates for all *PGK*(synEPOR) conditions; $N$ = 4 biological replicates for all *EPOR*(synEPOR) conditions; and $N$ = 3 biological replicates for all Mock conditions. **C** Percentage of total cell proliferation normalized to clones cultured +EPO over the course of differentiation. Bars represent mean

+/SEM; ns = not statistically significant, *$p$ = 0.0229, **$p$ = 0.0168, ***$p$ = 0.00193, ****$p$ = 0.000569, and *****$p$ < 0.00001 comparing +BB to +EPO conditions by unpaired two-tailed $t$ test across distinct samples. $N$ = 6 biological replicates for all *PGK*(synEPOR) conditions; $N$ = 4 biological replicates for all *EPOR*(synEPOR) conditions and Mock condition −BB/−EPO; and $N$ = 3 biological replicates for Mock conditions −BB/−EPO and +BB/-EPO. **D** Representative hemoglobin tetramer HPLC plots of edited and unedited iPSC-derived erythroid cells at end of differentiation. Delta mV was calculated by subtracting background mV value. **E** Ratio of HbF production in +BB v. +EPO conditions of synEPOR-edited iPSC-derived erythroid cells at end of differentiation. Source data are provided as a Source Data file. **F** Cost comparison of EPO and BB (lowest price per mg commercially available for purchase as of 2/9/24).

edited conditions, with clones cultured with BB typically producing elevated levels of fetal hemoglobin relative to those same clones cultured with EPO (Fig. 5D, E). However, there appeared to be some clonal variation since not all clones conformed to these trends (Fig. 5E). These findings were further confirmed by quantifying hemoglobin production per cell, which was done using HPLC to quantify the amount of heme released by hemoglobin based on a standard curve. This analysis revealed generally elevated hemoglobin production across *EPOR*(synEPOR)-edited clones cultured with BB compared to the same clones cultured with EPO (median of 33.1 vs. 24.4 pg hemoglobin per cell, respectively; Supplementary Fig. 16C). In contrast, clones edited with *PGK*(synEPOR) showed higher hemoglobin production when cultured with EPO (median of 21.1 vs. 32.1 pg hemoglobin per cell with BB vs. EPO, respectively). Importantly, these levels of hemoglobin production are within the range expected for normal RBCs in the blood stream (25.4–34.6 pg/cell)[34,35], with fetal hemoglobin being the predominant hemoglobin as reported using current iPSC differentiation protocols[33,36]. Finally, while transfused RBCs typically produce more adult than fetal hemoglobin, the healthy phenotype of patients with hereditary persistence of fetal hemoglobin (HPFH) and the recent approval of Casgevy to induce high levels of HbF to treat sickle cell disease and β-thalassemia provide support that a blood product with high HbF should be both safe and effective[37,38].

## Discussion

In this work, we combined synthetic protein engineering with the specificity of homology-directed repair genome editing to enable small molecule control of cell differentiation and behavior. By first optimizing highly effective small molecule-responsive receptors and then integrating them into endogenous regulatory machinery, we effectively recapitulated native receptor signaling. These efforts enable cell signaling to be stimulated by low-cost small molecules instead of recombinant cytokines currently required for ex vivo cell manufacturing. In this specific instance, EPO is one of the most expensive components of erythroid-promoting media[7]. Here, we demonstrate that *EPOR*(synEPOR)-edited cells cultured with a small molecule are capable of achieving equivalent erythroid differentiation, transcriptomic changes, and hemoglobin production compared to cells cultured with EPO. For comparison, we determined the cost per mg of the largest commercially available units of recombinant human EPO and AP20187 (BB) small molecule. We found that the price per mg of BB was nearly 50-fold less than that of recombinant EPO (Fig. 5F). In addition, 1/10th the amount of BB compared to recombinant EPO was required to yield equivalent erythroid production from *EPOR*(synEPOR)-edited iPSCs. Taken together, the corresponding estimates for cost of EPO required to produce a single unit of RBCs at a culture density of 5e7/mL is $1,246.50, conversely the cost to produce an equivalent amount of RBCs using BB is $2.25.

While the tools and editing strategies defined in the work enable the replacement of recombinant EPO with low-cost small molecules, the cost of EPO-free media still greatly exceeds the cost of donated blood[7]. However, there are instances where donated blood is not readily available, such as for those with exceptionally rare blood types, or among patients with chronic diseases that require repeated transfusions, such as sickle cell disease, where minor differences in antigen prevalence lead to the development of alloimmunization in up to 30% of patients[3]. Therefore, our technology could be integrated into patient-derived iPSCs to produce a renewable supply of autologous RBCs at reduced cost for these currently unmet medical needs. Yet, there are major advances still required before this approach will be biologically and economically feasible. These include further reducing the cost of erythroid-promoting media, better replicating high-density RBC production that occurs in vivo[33,39], and improving enucleation of adult hemoglobin-producing RBCs[40]. This is a multi-faceted problem and will require sustained efforts to further reduce production expenses. Nevertheless, by combining protein engineering and genome engineering to eliminate the requirement of EPO cytokine in erythroid-promoting media, this work demonstrates how synthetic biology may be used in the future to eliminate the need for other exogenous cytokines currently required for efficient ex vivo RBC production. For instance, orthogonal dimerizer systems could be leveraged in the future to create small molecule-inducible MPL, KIT, IL3R, and INSR to remove the need for TPO, SCF, IL-3, and insulin, respectively. We therefore believe this work brings us one major step closer to establishing ex vivo RBC production as a scalable and renewable source of blood cells for transfusion medicine.

More broadly, we envision a future where clinically relevant cell types may be manufactured off-the-shelf and at scale to meet the broad spectrum of patient needs. However, significant advances are needed to improve affordability and accessibility to patients. Given the complexities of large-scale cell manufacturing, many innovations have been accomplished by mechanical engineers who have developed improved bioreactors[33,41,42]. Our work demonstrates how challenges within this space may also be addressed by genome engineers to create more effective producer cells to seed these advanced bioreactors. Because dependence on expensive cytokines is a common barrier to scalable production of any cells ex vivo, the strategies defined in this work may be readily adapted to enable large-scale production of platelets, neutrophils, T cells, and many other clinically relevant cell types. This will ensure that advancements in cell engineering may be rapidly translated to patients at a cost that is both affordable and accessible.

Finally, this work demonstrates the power of iterative design-build-test cycles to rapidly improve the function of synthetic proteins. In this work, test cycle 1 defined the ideal placement of an FKBP domain within the EPO receptor. Test cycle 2 enhanced the efficacy of synEPOR designs by incorporation of signal peptides and a naturally occurring EPOR mutation. Finally, test cycle 3 defined the ideal expression profile of our optimized synEPOR cassette when placed under a variety of exogenous and endogenous promoters. Perhaps unsurprisingly, we find that integration of the optimized synEPOR at the endogenous *EPOR* locus best recapitulates endogenous EPOR signaling, an engineering attribute enabled by homology-directed repair genome editing. In addition, given the incredible modularity of membrane-bound receptors[43], it is possible that the small molecule-inducible architecture defined in this work may inform the design of other potentially useful small molecule-inducible receptors to modulate a wide variety of cell signals. If so, it is likely that synthetic receptor function may be fine-tuned by design-build-test cycles as well as genome editing to mediate precise integration into the genome. Gaining precisely regulated and tunable control over cells will thus pave the way for increasingly sophisticated cell engineering applications.

## Methods

### Ethical statement

Informed patient consent was acquired, and patients were recruited in accordance with protocol number 33813, which was approved by the NHLBI Institutional Review Board. HSPCs derived from both male and female donors were included in this study with no exclusion criteria applied regarding ethnicity. All samples were de-identified immediately following collection.

### Statistics and reproducibility

No statistical method was used to predetermine sample size. However, we conducted experiments with sample sizes sufficient to determine statistically significant differences across treatments. No data were excluded from the analyses. The experiments were not randomized, and investigators were not blinded to allocation during experiments or outcome assessment. All statistical tests on experimental groups were performed using GraphPad Prism software (v9).

## Integration vector design

Integration vectors were designed such that the left and right homology arms (LHA and RHA, respectively) are immediately flanking the cut site in exon 2 of the *CCR5* locus or exon 8 of the *EPOR* locus. For *HBA1* integration, full gene replacement was achieved using split homology arms—the LHA corresponding to the region immediately upstream of the start codon and RHA corresponding to the region immediately downstream of the cut site in the 3′ UTR of the *HBA1* gene[10]. Homology arm length ranged from 400–1000 bp. For FKBP-EPOR chimeras, flexible GGGGS linkers were added between FKBP domains and SPs and the *EPOR* gene. When placing the FKBP domain immediately adjacent to the EPOR transmembrane domain, the TM domain was defined as amino acid sequence PLILTLSLILVVILVLLTVLALLSH. EPOR SP was defined as amino acid sequence MDHLGASLWPQVGSLCLLLA-GAAW. IL6 SP was defined as amino acid sequence MNSFSTSAFGP-VAFSLGLLLVLPAAFPAP. The FKBP corresponded to amino acid sequence MLEGVQVETISPGDGRTFPKRGQTCVVHYTGMLEDGKKVDSS RDRNKPFKFMLGKQEVIRGWEEGVAQMSVGQRAKLTISPDYAYGATGHP GIIPPHATLVFDVELLKLE. Finally, to avoid the possibility of unintended recombination of synEPOR with the endogenous locus for *EPOR*(synEPOR)-edited conditions, we disguised homology of synEPOR by creating silent mutations within the EPOR domains at every possible codon, with a preference for codons that occurred more frequently throughout the human genome[44]. All custom sequences for cloning were ordered from Integrated DNA Technologies (IDT; Coralville, Iowa, USA). Gibson Assembly MasterMix (New England Biolabs, Ipswich, MA, USA) was used for the creation of each vector as per the manufacturer's instructions.

## AAV6 DNA repair template production and purification

All AAV6 vectors were cloned into the pAAV-MCS plasmid (Agilent Technologies, Hayward, CA, USA), which contains inverted terminal repeats (ITRs) derived from AAV2. To produce AAV6 vectors, we seeded HEK293T cells (CRL-1573, ATCC, Manassas, VA, USA) in 2-5 15 cm² dishes at 13-15×10⁶ cells per plate; 24 h later, each dish was transfected using 112 μg polyethylenemine (cat.: 23966; Polysciences, Warrington, PA, USA), 6 μg of ITR-containing plasmid, and 22 μg of pDGM6 (gift from D. Russell, University of Washington), which contains the AAV6 cap genes, AAV2 rep genes, and Ad5 helper genes. After 48–72 h of incubation, cells were collected and AAV6 capsids were isolated using the AAVPro Purification Kit (All Serotypes, Takara Bio, San Jose, USA), as per the manufacturer's instructions. AAV6 vectors were titered using a Bio-Rad QX200 ddPCR machine and QuantaSoft software (v.1.7, Bio-Rad, Hercules, CA, USA) to measure the number of vector genomes.

## HSPC culture

CD34⁺ HSPCs were sourced from fresh cord blood (generously provided by the Stanford Binns Family Program for Cord Blood Research) and Plerixafor- and/or G-CSF-mobilized peripheral blood (AllCells, Alameda, CA, USA or STEMCELL Technologies, Vancouver, Canada). CD34⁺ HSPCs were cultured at 1-5×10⁵ cells/mL in StemSpan SFEMII (STEMCELL Technologies) or Good Manufacturing Practice Stem Cell Growth Medium (SCGM, CellGenix, Freiburg, Germany) base medium supplemented with a human cytokine (PeproTech, Rocky Hill, NJ, USA) cocktail: stem cell factor (100 ng/mL), thrombopoietin (100 ng/mL), Fms-like tyrosine kinase 3 ligand (100 ng/ml), interleukin-6 (100 ng/mL), streptomycin (20 mg/mL) (ThermoFisher Scientific, Waltham, MA, USA), and penicillin (20 U/mL) (ThermoFisher Scientific, Waltham, MA, USA), and 35 nM of UM171 (cat.: A89505; APExBIO, Houston, TX, USA). The cell incubator conditions were 37 °C, 5% $CO_2$, and 5% $O_2$.

## Genome editing of HSPCs

Chemically modified CRISPR guide RNAs (gRNAs) used to edit CD34⁺ HSPCs at *CCR5*, *HBA1*, and *EPOR* were purchased from Synthego (Redwood City, CA, USA). The gRNA modifications added were 2′-O-methyl-3′-phosphorothioate at the three terminal nucleotides of the 5′ and 3′ ends[45]. The target sequences for gRNAs were as follows: *CCR5*: 5′-GCAGCATAGTGAGCCCAGAA-3′; *HBA1*: 5′-GGCAAGAAGCATGGCCA CCGAGG-3′; and *EPOR*: 5′-AGCTCAGGGCACAGTGTCCA-3′. All Cas9 protein was purchased from Aldevron (Alt-R S.p. Cas9 Nuclease V3; Fargo, ND, USA). Cas9 ribonucleoprotein (RNP) complexes were created at a Cas9:gRNA molar ratio of 1:2.5 at 25 °C for a minimum of 10 min before electroporation. CD34⁺ cells were resuspended in P3 buffer plus supplement (cat.: V4XP-3032; Lonza Bioscience, Walkersville, MD, USA) with complexed RNPs and electroporated using the Lonza 4D Nucleofector (program DZ-100). Cells were plated at 1-2.5×10⁵ cells/mL following electroporation in the cytokine-supplemented media described above. Immediately following electroporation, AAV6 was supplied to the cells between 2.5-5e3 vector genomes per cell. The small molecule AZD-7648, a DNA-dependent protein kinase catalytic subunit inhibitor, was also added to cells immediately post-editing for 24 h at 0.5 nM to improve homology-directed repair frequencies[46].

## Ex vivo erythroid differentiation

Following editing, HSPCs derived from healthy patients or iPSC-derived HPCs were cultured for 14 d at 37 °C and 5% $CO_2$ in SFEMII medium (STEMCELL Technologies, Vancouver, Canada)[21,22]. SFEMII base medium was supplemented with 100 U/mL penicillin/streptomycin (ThermoFisher Scientific, Waltham, MA, USA), 10 ng/mL stem cell factor (PeproTech, Rocky Hill, NJ, USA), 1 ng/mL interleukin-3 (PeproTech, Rocky Hill, NJ, USA), 3 U/mL EPO (eBiosciences, San Diego, CA, USA), 200 μg/mL transferrin (Sigma-Aldrich, St. Louis, MO, USA), 3% antibody serum (heat-inactivated; Sigma-Aldrich), 2% human plasma (isolated from umbilical cord blood provided by Stanford Binns Cord Blood Program), 10 μg/mL insulin (Sigma-Aldrich), and 3 U/mL heparin (Sigma-Aldrich). In the first phase, at days 0–7 of differentiation (d0 being 2–3 d post editing), cells were cultured at 1×10⁵ cells/mL. In the second phase (d7–10), cells were maintained at 1×10⁵ cells/mL and IL-3 was removed from the culture. In the third phase (d11–14), cells were cultured at 1×10⁶ cells/mL and transferrin was increased to 1 mg/mL. For -EPO conditions, cells were cultured in the same culture medium listed above except for removal of EPO from the media. For conditions with the addition of BB homodimerizer (AP20187; Takara Bio, San Jose, USA), 1 μL of 0.5 mM BB was diluted in 999 μL PBS (HI30; BD Biosciences, San Jose, CA, USA), of which 2 μL of the dilution was added for every 1 mL of differentiation media to reach a desired concentration of 1 nM. Fresh BB was added at each media change (d0, 4, 7, 11). For experiments requiring additional dilutions, BB was diluted further in PBS to reach the required concentration (as low as 1pM). For conditions with EPO mimetic, EMP17 (Anaspec, Fremont, CA, USA) was added to differentiation media at a molar ratio equivalent to 3 U/mL of EPO (618pM) in place of recombinant EPO.

## Immunophenotyping of differentiated erythrocytes

HSPCs subjected to erythroid differentiation were analyzed at d14 for erythrocyte lineage-specific markers using a FACS Aria II and FACS Diva software (v.8.0.3; BD Biosciences, San Jose, CA, USA). Edited and unedited cells were analyzed by flow cytometry using the following antibodies: CD34-APC (1:50 dilution; 561; BioLegend, San Diego, CA, USA), CD45-V450 (1:50 dilution; 2 μL in 100 μl of pelleted RBCs in 1×PBS buffer; HI30; BD Biosciences), CD36-PE (1:50 dilution; 5–271; BioLegend), CD71-PE-Cy7 (1:500 dilution; OKT9; Affymetrix, Santa Clara, CA, USA), and CD235a (GPA)-PE (1:500 dilution; GA-R2; BD Biosciences) or GPA-PE-Cy5 (1:500 dilution; GA-R2; BD Biosciences). In addition to cell-specific markers, cells were also stained with Ghost Dye Red 780 (Tonbo Biosciences, San Diego, CA, USA) to measure viability and DRAQ5 to quantify enucleation frequencies (BioLegend). All data visualization was performed using the FACS Aria II cytometer and

FACS Diva software (v.8.0.3) and subsequent data analysis was performed using FlowJo (v.10.6.1).

## Editing frequency analysis

Between 2–4 d post editing, HSPCs were harvested and QuickExtract DNA extraction solution (Epicentre, Madison, WI, USA) was used to collect genomic DNA (gDNA). Additional samples were collected at various stages of erythroid differentiation (d4, 7, 11, and 14) and gDNA was digested using BamHI-HF as per the manufacturer's instructions (New England Biolabs, Ipswich, MA, USA). Percentage of targeted alleles within a cell population was measured with a Bio-Rad QX200 ddPCR machine and QuantaSoft software (v.1.7; Bio-Rad, Hercules, CA, USA) using the following reaction mixture: 1-4 µL of digested gDNA input, 10 µL of ddPCR SuperMix for Probes (no dUTP) (Bio-Rad), primer/probes (1:3.6 ratio; Integrated DNA Technologies, Coralville, IA, USA) and volume up to 20 µL with $H_2O$. ddPCR droplets were then generated following the manufacturer's instructions (Bio-Rad): 20 µL of ddPCR reaction, 70 µL of droplet generation oil, and 40 µL of droplet sample. Thermocycler (Bio-Rad) settings were as follows: 98 °C (10 min), 94 °C (30 s), 57.3 °C (30 s), 72 °C (1.75 min), return to step 2 × 40-50 cycles, and 98 °C (10 min). Analysis of droplet samples was performed using the QX200 Droplet Digital PCR System (Bio-Rad). To determine percentages of alleles targeted, the numbers of Poisson-corrected integrant copies/mL were divided by the numbers of reference DNA copies/mL. The following primers and 6-FAM/ZEN/IBFQ-labeled hydrolysis probes were purchased as custom-designed PrimeTime quantitative PCR (qPCR) assays from Integrated DNA Technologies: All *HBA1* vectors: forward: 5′-AGTCCAAGCTGAGCAAAGA-3′, reverse: 5′-ATCACAAACGCAGGCAGAG-3′, probe: 5′-CGAGAAGCGC GATCACATGGTCCTGC-3′; all *CCR5* vectors: forward: 5′-GGGAG-GATTGGGAAGACAAT-3′, reverse: 5′-TGTAGGGAGCCCAGAAGAGA-3, probe: 5′-CACAGGGCTGTGAGGCTTAT-3′. The primers and HEX/ZEN/IBFQ-labeled hydrolysis probe, purchased as custom-designed PrimeTime qPCR Assays from Integrated DNA Technologies, were used to amplify the *CCRL2* reference gene: forward: 5′-GCTGTATGAATC-CAGGTCC-3′, reverse: 5′-CCTCCTGGCTGAGAAAAAG-3′, probe: 5′-TGTTTCCTCCAGGATAAGGCAGCTGT-3′.

## Cresyl blue staining

On d14 of RBC differentiation, up to 1e4 cells were loaded on a glass slide in no more than 5 µL volume of PBS. Then 9 µL of Brilliant Cresyl blue staining solution (cat.: #16035; Sigma-Aldrich) was added onto a coverslip and allowed to air dry. The coverslip was then placed over the cells on the glass slide to stain for reticulocytes. Brightfield images were taken at ×20 magnification.

## AlphaFold2 structural predictions

Energy-predicted structures were derived by applying AlphaFold2 (v2.3.2)[23] on wild-type EPOR, truncated EPOR, and synEPOR sequences. Five differently trained neural networks were applied to produce unrelaxed structure predictions. Energy minimization was applied to the best predicted unrelaxed structure (highest average predicted distance difference test (pLDDT) and lowest predicted aligned error) to produce the optimal relaxed structure.

## Hemoglobin tetramer analysis

Frozen pellets of approximately 1e6 cells ex vivo-differentiated erythroid cells were thawed and lysed in 30 µL of RIPA buffer with 1x Halt Protease Inhibitor Cocktail (ThermoFisher Scientific, Waltham, MA, USA) for 5 min on ice. The mixture was vigorously vortexed and cell debris was removed by centrifugation at 20,700 g for 10 min at 4 °C. HPLC analysis of hemoglobins in their native form was performed on a cation-exchange PolyCAT A column (35 × 4.6mm², 3 µm, 1500 Å; PolyLC Inc., Columbia, MD, USA) using a Perkin-Elmer Flexar HPLC system (Perkin-Elmer, Waltham, MA, USA) at room temperature and

detection at 415 nm. Mobile phase A consisted of 20 mM Bis-tris and 2 mM KCN at pH 6.94, adjusted with HCl. Mobile phase B consisted of 20 mM Bis-tris, 2 mM KCN, and 200 mM NaCl at pH 6.55. Hemolysate was diluted in buffer A prior to injection of 20 µL onto the column with 8% buffer B and eluted at a flow rate of 2 mL/min with a gradient made to 40% B in 6 min, increased to 100% B in 1.5 min, returned to 8% B in 1 min, and equilibrated for 3.5 min. Quantification of the area under the curve of peaks was performed with TotalChrom software (Perkin-Elmer) and raw values were exported to GraphPad Prism v9 software for plotting and further analysis.

## Bulk RNA-sequencing

Total RNA was extracted from frozen pellets of approximately 1e6 cells per condition using RNeasy Plus Micro Kit (Qiagen, Redwood City, CA, USA) according to the manufacturer's instructions. Sequencing was provided by Novogene (Sacramento, CA, USA) and raw FASTQ files were aligned to the GRCh38 reference genome extended with the *synEPOR* target sequence and quantified using Salmon (v1.9.0)[47] with default parameters. Quality control was performed by Novogene.

## Differential gene expression analysis and gene set enrichment analysis

The estimated gene expression counts were used with DESeq2[48] to conduct differential gene expression analysis between sample groups. Mitochondrial and lowly expressed genes were removed (sum NumReads <1). The top 50 up- and down-regulated genes based on adjusted p-value using Wald test were isolated and analyzed with Enrichr[49] to yield functional annotations.

## Principal component analysis and gene distribution plots

Mitochondrial genes were removed from the gene expression matrix (TPM) and the remaining genes were used to conduct principal component analysis with all samples. Gene expression for experimental and control groups were averaged and log-normalized. Average gene expression distributions were plotted using Seaborn (https://github.com/atsumiando/RNAseq_figure_plotter_python).

## Gene co-expression network analysis

The TPM-normalized gene expression matrix of all *PGK*(synEPOR)-, *HBA1*(synEPOR)-, and *EPOR*(synEPOR)-edited conditions (n = 10) was used to construct a pairwise gene similarity matrix where each entry represented the Spearman correlation coefficient between a pair of genes. The correlation between a specified set of *EPOR*-related genes was compared for both *EPOR* and *synEPOR* to determine which genes *synEPOR* adequately mimics in the immediate gene co-expression network of endogenous *EPOR*.

## iPSC line and culture

A previously published iPSC line, PB005 derived from peripheral blood of a donor with O⁻ blood type was used in this study[50]. iPSCs were cultured and maintained in mTeSR1 medium (cat.: 85850; STEMCELL Technologies, Vancouver, Canada) on Matrigel (cat.: 354277; Corning, NY, USA)-coated plates. For passaging, cells at a confluency of 80-90% were incubated with Accutase (cat.: AT104; Innovative Cell Technologies, San Diego, USA) for 5–7 min to dissociate into single cells and replated in mTeSR1 medium supplemented with 10 mM of ROCKi (Y27632; cat.: 10005583; Cayman Chemical, Ann Arbor, MI, USA). After 24 h, cells were maintained in fresh mTeSR1 medium with daily media changes. For freezing iPSCs, STEM-CELLBANKER freezing medium (cat.: 11924; Amsbio, Cambridge, MA, USA) was used.

## Genome editing of iPSCs

iPSCs were genome edited using the CRISPR/AAV platform[19,46] as follows: Cas9 RNP complex was formed by combining 5 µg of Cas9 (Alt-R S.p. Cas9 Nuclease V3; Fargo, ND, USA) and 2 µg of gRNA (Synthego,

Redwood City, CA, USA) and incubating at room temperature for 15 min. iPSCs pre-treated with ROCKi (Y27632; cat.: 10005583; Cayman Chemical, Ann Arbor, MI, USA) for 24 h were dissociated with Accutase (cat.: AT104; Innovative Cell Technologies, San Diego, USA) into single cells. 1-5e5 iPSCs were resuspended in 20 µL of P3 primary cell nucleofector solution plus supplement (cat.: V4XP-3032; Lonza Bioscience, Walkersville, MD, USA) along with the RNP complex and electroporated using Lonza 4D Nucleofector (program CA-137). After electroporation, iPSCs were plated in mTeSR1 medium supplemented with ROCKi, 0.25 µM AZD7648 (cat.: S8843; Selleck Chemicals, Houston, TX, USA) and AAV6 donor at 2.5e3 vector genomes per cell, based on ddPCR titers as above. After 24 h, cells were switched to medium with mTeSR1 and ROCKi. From the following day, cells were maintained in mTeSR1 medium without ROCKi.

### Single-cell cloning of iPSCs

To isolate single-cell'' clones, genome-edited iPSCs were plated at a density of 250 cells per well of a 6-well plate in mTeSR1 medium supplemented with 1x CloneR2 reagent (cat.: 100-0691; STEMCELL Technologies, Vancouver, Canada). After 48 h, cells were switched to fresh mTeSR1 medium with 1x CloneR2 and incubated for 2 d. Following this, iPSCs were maintained in mTeSR1 medium without CloneR2 with daily media changes. At d7–10, single-cell colonies were picked by scraping and propagated individually. The isolated single cell iPSCs were genotyped using PCR with primers annealing outside the homology arms to identify clones with bi-allelic knock-in. The following primers were used for genotyping: *CCR5* integration: forward: 5′-CTCA-TAGTGCATGTTCTTTGTGGGC-3′, reverse: 5′-CCAGCCCAGGCTGTG-TATGAAA-3′; *EPOR* integration: forward: 5′-GCCACATGGCTAGAG TGGTAT-3′, reverse: 5′-CTTTCTTAGAACATGGCCTGATTCAGA-3′.

### iPSC-to-erythrocyte differentiation

iPSCs were differentiated into CD34+ HPCs using the STEMdiff Hematopoietic Kit (cat.: 05310; STEMCELL Technologies, Vancouver, Canada) according to the manufacturer's protocol. Briefly, iPSCs at 70-80% confluency were dissociated into aggregates using ReLeSR (cat.: 100-0484; STEMCELL Technologies). Aggregates were then diluted 10-fold, and 100 µL of the diluted suspension was aliquoted into a 96-well plate for quantification. Approximately 80 aggregate colonies were subsequently plated per well of a 12-well plate pre-coated with Matrigel and maintained in mTeSR1 medium. 24 h post-plating, the number of colonies per well was manually quantified, and the medium was replaced with differentiation medium A. The medium was then changed according to the kit's instructions for a total of 12 days. On d12, suspension cells were harvested by pipetting cells up and down to ensure a homogeneous cell suspension. To assess the efficiency of differentiation, as determined by CD34+/CD45+ expression, cells were analyzed using flow cytometry with the erythrocyte flow panel described above for HSPCs. Following this, CD34+ cells were further differentiated into erythroid cells using the three-phase system described above, either in the presence or absence of EPO and BB.

### Heme detection analysis

Quantification of the amount of hemoglobin produced in cells was obtained by quantitative detection of the heme peak released from hemoglobin. Lysate were obtained from 1-2e5 cells as frozen pellets, as described for hemoglobin tetramer analysis. The relationship between heme and hemoglobin was established from serially diluted hemolysate made with a blood sample of a known hemoglobin content. Detection of heme was performed by reverse-phase PerkinElmer Flexar HPLC system (PerkinElmer) with a Symmetry C18 column (4.6 × 75 mm, 3.5 µm; Waters Corporation, Milford, MA, USA) at 415 nm. Mobile phase A consisted of 10% methanol made in acetonitrile and

mobile B of 0.5% trifluoroacetic acid in water adjusted at pH 2.9 with NaOH. Samples were injected at a flow rate of 2 mL/min in 49% A, followed by a 3 min gradient to 100% A. The column was then equilibrated to 49% A for 3 min.

### Reporting summary

Further information on research design is available in the Nature Portfolio Reporting Summary linked to this article.

## Data availability

RNA-seq data have been deposited in the NCBI Gene Expression Omnibus database (accession no. GSE285656). Sequencing reads were aligned to the GRCh38 reference human genome (NCBI Sequence Read Archive database; accession no. PRJNA31257). The data for all figures in this study are provided in the Source Data file. Source data are provided with this paper.

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

## Acknowledgements

The authors thank the following funding sources that made this work possible: the Stanford University Knight-Hennessy Scholarship, the Paul & Daisy Soros Fellowship for New Americans, and the Hertz Fellowship to F.K.E.; Stanford University Medical Scholars Research Program, the American Society of Hematology Minority Medical Student Award Program, and the Stanford University Medical Scientist Training Program to S.E.L.; California Institute for Regenerative Medicine Bridges Program for T.M. and T.T.; National Science Foundation Graduate Research Fellowship Program to B.J.L.; NIH T32 Research Training in Transplant Surgery Fellowship (1T32AI125222-01) to S.N.C.; University of California, San Francisco Program for Breakthrough Biomedical Research: New Frontier Research Award to M.K.C; and Mary Anne Koda-Kimble Seed Award for Innovation to M.K.C. We also would also like to thank the Stanford Binns Program for Cord Blood Research for providing CD34[+] HSPCs and the FACS Core Facility at the Stanford Institute of Stem Cell Biology and Regenerative Medicine as well as University of California, San Francisco Flow Core for access to flow cytometry machines. Finally, we would like to thank Caleb Grossman for helpful discussions and feedback as we planned initial experiments.

## Author contributions

M.H.P. and M.K.C. supervised the project. A.P.S., K.R.M., C.T.C., M.H.P., and M.K.C. designed experiments. A.P.S., K.R.M., F.K.E., S.S., D.S., R.S., E.S., S.E.L., T.M., B.J.L., T.T., S.N.C., and M.K.C. carried out experiments. A.P.S., K.R.M., F.K.E., P.C., M.H.P., and M.K.C. analyzed data. A.P.S., K.R.M., and M.K.C. wrote the manuscript.

## Competing interests

M.H.P. is a member of the scientific advisory board of Allogene Therapeutics. M.H.P. has equity in CRISPR Tx and Kamau Tx. C.T.C., M.H.P., and M.K.C. have filed provisional patent no. PCT/US2023/076969, which includes all synEPOR designs, genomic integration strategies, and the application of EPO-independent erythropoiesis. The remaining authors declare no competing interests.
