## [Peer Review file · Nature Communications]

Engineering synthetic signaling receptors to enable erythropoietin-free erythropoiesis

Corresponding Author: Dr M. Cromer

Version 0:

Reviewer comments:

Reviewer #1

(Remarks to the Author)

The authors did a good job addressing my comments. I don't have any other critiques.

Reviewer #4

(Remarks to the Author)

In this revised version the authors properly addressed the most relevant critiques raised by the reviewer, resulting in a significantly improved manuscript.

My only minor concern relates to the lack of clarity of graph legends: in detail, while it is clear that +Epo and +BB indicate samples treated with each single agent, it is not immediate to understand what +/- and -/- mean. I would suggest to include a description in the figure legend. Also, in Figure S14C the annotation of the x axis is missing.

Dear Nature Communications Editorial Staff,

We appreciate the time and effort that you and the reviewers have dedicated to providing your valuable feedback on our manuscript. We have incorporated changes to reflect the suggestions provided by all reviewers and have itemized a point-by-point response to the additional comments and suggestions.

Reviewer #1 (Remarks to the Author):

The authors did a good job addressing my comments. I don't have any other critiques.

We appreciate the positive feedback. We again thank the reviewers for their insightful comments which have helped further improve our manuscript.

Reviewer #4 (Remarks to the Author):

In this revised version the authors properly addressed the most relevant critiques raised by the reviewer, resulting in a significantly improved manuscript.

My only minor concern relates to the lack of clarity of graph legends: in detail, while it is clear that +Epo and +BB indicate samples treated with each single agent, it is not immediate to understand what ++ and -- mean. I would suggest to include a description in the figure legend. Also, in Figure S14C the annotation of the x axis is missing.

This has been corrected.